# Preparation of 3D Nd_2_O_3_-NiSe-Modified Nitrogen-Doped Carbon and Its Electrocatalytic Oxidation of Methanol and Urea

**DOI:** 10.3390/nano13050814

**Published:** 2023-02-22

**Authors:** Simin Zhang, Ying Chang, Aiju Xu, Jingchun Jia, Meilin Jia

**Affiliations:** Inner Mongolia Key Laboratory of Green Catalysis and Inner Mongolia Collaborative Innovation Center for Water Environment Safety, College of Chemistry and Environmental Science, Inner Mongolia Normal University, Hohhot 010022, China

**Keywords:** methanol oxidation reaction, urea oxidation reaction, rare earth, nickel-based catalyst

## Abstract

**Simple Summary:**

It is vitally important that scientists are able to describe their work simply and concisely to the public, especially in an open access online journal. The simple summary consists of no more than 200 words in one paragraph and contains a clear statement of the problem addressed, the aims and objectives, pertinent results, conclusions from the study, and how they will be valuable to society. This should be written for a lay audience, i.e., no technical terms without explanations. No references are cited, and no abbreviations included. Submissions without a simple summary will be returned directly.

**Abstract:**

Developing renewable energy sources and controlling water pollution are critical but challenging problems. Urea oxidation (UOR) and methanol oxidation (MOR), both of which have high research value, have the potential to effectively address wastewater pollution and energy crisis problems. A three-dimensional neodymium-dioxide/nickel-selenide-modified nitrogen-doped carbon nanosheet (Nd_2_O_3_-NiSe-NC) catalyst is prepared in this study by using mixed freeze-drying, salt-template-assisted technology, and high-temperature pyrolysis. The Nd_2_O_3_-NiSe-NC electrode showed good catalytic activity for MOR (peak current density ~145.04 mA cm^−2^ and low oxidation potential ~1.33 V) and UOR (peak current density ~100.68 mA cm^−2^ and low oxidation potential ~1.32 V); the catalyst has excellent MOR and UOR characteristics. The electrochemical reaction activity and the electron transfer rate increased because of selenide and carbon doping. Moreover, the synergistic action of neodymium oxide doping, nickel selenide, and the oxygen vacancy generated at the interface can adjust the electronic structure. The doping of rare-earth-metal oxides can also effectively adjust the electronic density of nickel selenide, allowing it to act as a cocatalyst, thus improving the catalytic activity in the UOR and MOR processes. The optimal UOR and MOR properties are achieved by adjusting the catalyst ratio and carbonization temperature. This experiment presents a straightforward synthetic method for creating a new rare-earth-based composite catalyst.

## 1. Introduction

Because of the increasing concern regarding environmental pollution and the depletion of fossil fuels, developing renewable energy techniques and controlling water pollution are crucial [1]. Hydrogen has always been considered as a clean and sustainable source of energy owing to its abundant reserves, high specific gravity [2], and energy density, making it renewable, recyclable [3], and an ideal alternative energy source [4,5]. Hydrogen production, by employing electrochemical decomposition and hydrolysis as well as fuel cell fabrication, can solve the energy crisis while meeting the goals of green chemistry [6,7,8]. At present, a widely studied electrochemical hydrogen production method is the combination of the hydrogen evolution reaction (HER) and oxygen evolution reaction (OER) [9,10,11]. This method incurs a high total cost, mainly because it relies on Pt as a result of the lack of an effective HER catalyst under alkaline conditions. Therefore, more energy-efficient hydrogen production methods, such as methanol oxidation or urea oxidation, which can be used to replace the traditional OER, must be studied. Methanol has a simple structure, is inexpensive, and can be directly used as an anode in a methanol fuel cell. It can also solve the problem of methanol-containing sewage treatment and provide unlimited possibilities for the development of new fuel cells [12,13]. Urea is the main component of both sewage and human/animal urine [14,15], and it can be electrochemically oxidized to N_2_. As the anode of a fuel cell, urea has three advantages: (1) the theoretical total voltage of urea electrolysis is lower than that of water electrolysis, reducing H_2_ production energy consumption; (2) harmful nitrides in the water system can be repaired; and (3) using urea oxidation (UOR) as an anode reaction in hydrogen production can avoid the mixing of O_2_ and H_2_ and ensure the safety of the electrolytic cell. Therefore, the OER can be replaced with methanol oxidation (MOR) and UOR to accelerate hydrogen production and reduce reaction costs [16,17,18,19].

Pt alloys have been considered as the best catalyst in recent years [20]. However, Pt-based catalysts are expensive and scarce, making them unsuitable for large-scale production. Thus, finding and developing non-noble metal catalysts with low cost and high catalytic activity is important. The best catalyst for UOR in an alkaline medium is nickel [21]. However, a pure nickel catalyst has the disadvantage of having few electroactive sites and is easily poisoned by carbon monoxide (CO). Therefore, in recent years, Ni-modified electrocatalysts, such as nano nickel [22], NiOOH [23,24], nickel hydroxide [25], Ni-Co bimetallic hydroxides [26], and nickel selenide, have been developed rapidly and are widely used. Because of fewer active sites on the surface and poor electrochemical stability, nickel selenide requires further investigation.

Because of their unique electronic structure and physicochemical properties, rare-earth elements have been widely used in many fields, particularly in the field of electrocatalysis [27]. Rare-earth elements have excellent catalytic activity because they have unfilled 4f orbitals in which 4f electrons can reside in 5d orbitals, which are transformed into valence electrons to promote the electron conduction in rare-earth metals [28]. As a result, transition metal groups doped with rare-earth metals are excellent electrocatalysts. Furthermore, the addition of rare-earth oxides can improve the activity and stability of nickel-based anodes [29]. Among the rare-earth elements, neodymium has a large specific surface area, nano size, electrochemical redox properties, and environmentally friendly properties along with high catalytic activity. According to previous research, neodymium can promote nickel-based catalysts to have excellent catalytic performance [30].

In this study, as a result of the salt-template and pyrolysis method, various rare-earth elements (such as Nd, Y, Gd, Sm, Pr, and La) were effectively compounded. A comparison of the electrochemical tests revealed that the three-dimensional neodymium-dioxide/nickel-selenide-modified nitrogen-doped carbon-nanosheet (Nd_2_O_3_-NiSe-NC) catalyst has efficient MOR and UOR performance. In addition, neodymium oxide has abundant oxygen vacancy. Neodymium oxide was introduced into nitrogen-doped carbon-supported nickel-selenide catalysts and applied to the electrocatalytic oxidation of methanol and urea. Under the synergistic action of neodymium oxide and nickel selenide, more defects can be obtained as active sites, showing excellent electrocatalytic properties of methanol and urea. In this paper, the effective combination of rare-earth elements and nickel selenide was studied using the salt-template and pyrolysis methods. Rare-earth-modified catalysts, neodymium oxide and nickel selenide, have synergistic effects on the catalytic activities of UOR and MOR. Therefore, this experiment can be used for the research and development of low-cost, high-activity MOR and UOR catalysts.

## 2. Experimental Section

### 2.1. Preparation of Catalyst

To prepare a mixed solution, we dissolved 0.6547 g of nickel nitrate hexahydrate (Shanghai, China), 1.0877 g of neodymium nitrate hexahydrate (Shanghai, China), 1.5 g of glucose (Tianjin, China), and 10.53 g of sodium chloride (Shanghai, China) in 60 mL of ultrapure water; then froze it with liquid nitrogen in a freeze dryer for 24 h until it is freeze-dried; and removed the freeze-dried powder. We then placed it in a tubular furnace and calcined it at 800 °C. The solid was then centrifuged and filtered using ultrapure water. The filtered solid was dried in a vacuum oven, with dicyandiamide added in the same proportion as the sample after drying, and then calcined for 3 h in an argon atmosphere at 800 °C. Finally, this is calcined for 2 h in an air atmosphere at 300 °C to obtain the precursor of Nd_2_O_3_-NiSe_2_-NC. Two porcelain boats were filled, one with the calcined precursor and the other with selenium powder. The ratio of selenium powder to Ni nanoparticles was 1:5. The selenium-powder-filled porcelain boat was placed at the tube furnace’s vent, inflated with argon, heated to 450 °C at a rate of 2 °C min^−1^, and carbonized for 2 h. The catalyst developed is known as Nd_2_O_3_-NiSe_2_-450°C-NC. For comparison, we used the same procedure to prepare Nd_2_O_3_-Ni_2_Se_3_-350°C-NC, Nd_2_O_3_-NiSe_2_-550°C-NC, Nd_2_O_3_-NiSe-650°C-NC, and Nd_2_O_3_-NiSe-750°C-NC catalysts. We also prepared a comparison sample of the NiSe_2_-450°C-NC catalyst. This same procedure was used to create other rare-earth-element catalysts.

### 2.2. Physical Property Characterization

The German Bruker D8 advance device (Luken City, German) was used to perform powder X-ray diffraction (XRD) studies. The electronic structure of components in samples was studied using X-ray photoelectron spectroscopy (XPS, escalab 250xi, Thermo Fisher Scientific, MA, USA). The morphology of the catalyst was studied by field-emission transmission electron microscopy and scanning electron microscopy (SEM, Hitachi cold field emission s4800, Hitachi, Japan) (FE-TEM, TECNAI G2 F20, Hillsborough, OR USA). The Fei TECNAI G2 F20 apparatus (Hillsborough, Oregon, USA) was utilized for element mapping and X-ray spectroscopy (EDS) investigation.

### 2.3. Electrochemical Characterization

Using a CHI 760E electrochemical workstation (Shanghai, China) to obtain electrochemical data, MOR and UOR tests were performed in a conventional three-electrode configuration. The working electrode was a 0.5 × 0.5 cm^2^ carbon paper, with a platinum wire as a counter electrode, and a Hg/HgO electrode as a reference electrode. The slurry was composed of a 2 mg catalyst sample and a 400 μL solution (includes Nafion, deionized water, and ethanol). Two times, a 25 μL slurry was applied evenly on the carbon paper and dried under infrared light. The reversible hydrogen electrodes were used to represent the measured MOR and UOR potentials (i.e., vs. RHE). The infrared correction was not carried out according to all experimental data. The performance of the catalyst was evaluated by linear sweep voltammetry (LSV) and cyclic voltammetry (CV) in the range of 0.2~0.8 V. Electrochemical impedance spectroscopy (EIS) was performed at a voltage of 0.45 V in the frequency range of 0.01–100 kHz. The electrolytes used in the experiment were either KOH, KOH/methanol, or KOH/urea.

## 3. Results and Discussion

### 3.1. Physical Characterizations

In this process of catalyst preparation, the precursor of nickel neodymium was prepared by mixed freeze-drying, salt-template-assisted technology, and high-temperature pyrolysis, and then dicyandiamide was added to the precursor. Dicyandiamide-assisted pyrolysis is used to provide a nitrogen source for the preparation of the catalyst. Finally, the Nd_2_O_3_-NiSe-NC catalyst was prepared by selenidation at various temperatures and was used in MOR and UOR. Figure 1 shows the preparation process of Nd_2_O_3_-NiSe_2_-450°C-NC.

SEM images of the catalyst before selenidation show the lamellar structure, but after selenidation, the catalyst showed a cluster structure due to the more aggregated substances. According to the comparison diagram before and after selenidation, the structure of the catalyst did not change significantly. This shows that selenidation will not destroy the catalyst’s morphology (Figure 2a,b). The TEM image (Figure 2c) shows the three-dimensional-sheet structure of the catalyst, which should be formed on the carbon skeleton substrate, while the HR-TEM (Figure 2d) shows the lattice structure. The sample surface is covered with uniform stripe spacings of 0.353 nm and 0.264 nm, which correspond to the (411) and (110) NiSe_2_ and Nd_2_O_3_ lattice surfaces, respectively. The mapping diagram (Figure 2e) shows that the catalyst contains six elements: C, N, Nd, O, Ni, and Se, which are evenly distributed.

This observation corroborated the XRD results. X-ray diffraction was used to examine the material’s diffraction pattern (XRD). The results show that the XRD of catalysts calcined at different temperatures has the same peak value as Nd_2_O_3_. These catalysts successfully formed rare-earth-metal oxides. The difference is that the nickel selenide has the same peak value as Ni_3_Se_2_ at 350 °C and NiSe_2_ when calcined at 450 °C and 550 °C (Figure 3a). At the same time, when the calcination temperature reaches 650 °C and 750 °C, the XRD diffraction peak corresponds to NiSe (Appendix A). The hexagonal NiSe, obtained by converting diamond Ni_3_Se_2_ to cubic NiSe_2_ and then heating above 650 °C, indicates that with an increase in temperature more selenium evaporates, resulting in a decrease in selenium content. In XRD, at temperatures above 650 °C, a diffraction peak of NdOCl was also observed. The surface element composition and chemical valence of the catalyst were further studied by X-ray photoelectron spectroscopy (XPS). XPS spectra of Nd_2_O_3_-NiSe_2_-450°C-NC showed C, N, Nd, O, Ni, and Se peaks, indicating the presence of these elements (Appendix A). Nd_2_O_3_-NiSe_2_-450°C-NC, NiSe_2_-NC, and Nd_2_O_3_-NC catalysts have three main components in the C1s spectrum (Appendix A): C–C (284.7 eV), C-N (286.0 eV), and O–C=O (289.3 eV) [31]. Significant changes in binding energies (BEs) were observed in high-resolution Ni2p and Se3d spectra. The Ni2p spectrum of Nd_2_O_3_-NiSe_2_-450°C-NC, as shown in Figure 3b, exhibits two contributions, namely Ni2p_3/2_ and Ni2p_1/2_ (the result of spin-orbit splitting), located at 855.2 and 873.1 eV, respectively, with satellite peaks at 861.6 and 879.6 eV. Similarly, it can be seen from the integral area of Ni^2+^ that Ni^2+^ and Ni^3+^ have changed. With the NiSe_2_-NC catalyst [32], after adding the oxide of rare-earth-metal neodymium, the BEs of Nd_2_O_3_-NiSe_2_-450°C-NC showed a positive displacement of 0.2 eV compared to NiSe_2_-NC. These significant changes demonstrate that when Nd_2_O_3_-NiSe_2_-450°C-NC is formed, Ni atoms exhibit an electron inflow trend compared to NiSe_2_-NC [33]. Similarly, when compared to a single component NiSe_2_-NC, the peak of Nd_2_O_3_-NiSe_2_-450°C-NC corresponding to Se3d_5/2_ and Se3d_3/2_ spin orbits shifted negatively by 0.2 eV; Figure 3c shows the tendency for electrons to flow out of Nd_2_O_3_-NiSe_2_-450°C-NC. These findings show that a strong electronic interaction is formed between Nd doping and NiSe, which jointly regulates the electronic structure of the generated Nd_2_O_3_-NiSe-NC and has a certain optimization for the subsequent oxidation performance of methanol and urea. The Nd3d spectrum (Figure 3d) shows that the catalyst has strong peaks, and the central binding energies corresponding to Nd3d_5/2_ and Nd3d_3/2_ are 982.2 eV and 1004.5 eV, respectively. The characteristic peak of Nd^3+^ is reflected in Nd_2_O_3_-NiSe_2_-450°C-NC and Nd_2_O_3_-NC catalysts [34]. As observed in the O1s spectrum (Figure 3e), the Nd_2_O_3_-NiSe_2_-450°C-NC and Nd_2_O_3_-NC catalysts contain adsorbed O (529.4 eV), vacancy O (531.5 eV), and Nd-O (532.21 eV) bonds. The NiSe-NC catalyst contains no Nd–O (532.21 eV) bond. In the N1s spectra (Figure 3f), the catalyst contains pyridine-N, pyrrole-N, and graphite-N [35].

### 3.2. Electrochemical Characterizations

The catalysts with MOR and UOR were tested to study the reaction behavior of various rare-earth-metal-modified nickel-based catalysts. Thus, all the rare-earth-metal-modified nickel-based catalysts have redox peaks, indicating a redox reaction between Ni^2+^ and Ni^3+^. According to the electrochemical test LSV curve, Nd_2_O_3_-NiSe_2_-450°C-NC has the highest initial potential when compared to other rare-earth-metal-modified nickel-based catalysts (Appendix A). This can be observed in the CV curves for methanol oxidation and urea oxidation (Appendix A). EIS tests on nickel-based catalysts of various rare-earth metals show that the spectrum of all samples is semicircular (Appendix A). The Tafel slopes of different rare-earth-doped catalysts for MOR and UOR were calculated by an LSV test (Appendix A). It showed that Nd_2_O_3_-NiSe_2_-450°C-NC has the smallest Tafel slope in both MOR and UOR tests, indicating that it has a fast kinetic process and that the catalyst is more prone to methanol oxidation and urea oxidation than other rare-earth-doped catalysts. The semicircle radius of Nd_2_O_3_-NiSe_2_-450°C-NC is significantly smaller than that of other rare-earth nickel-based catalysts, indicating that Nd_2_O_3_-NiSe_2_-450°C-NC has a low charge transfer resistance and faster electrocatalysis kinetics than other catalysts. The SEM images (Appendix A) of various rare-earth-metal-modified nickel-based catalysts were obtained to show different morphologies. Nickel-based catalysts doped with different rare earths at scan rate of 20, 40, 60, 80, and 100 mV s^−1^ in 1 M KOH solution (Appendix A) and the electrochemical active surface area (ECSA) was used as a reference for further understanding the activity of various rare-earth-metal nickel-based catalysts. C_dl_ was calculated by using the non-Faradaic process of CV curves of different catalysts. Nd_2_O_3_-NiSe_2_-450°C-NC has a higher C_dl_ than other rare-earth nickel-based catalysts (Appendix A). The CVs of Nd_2_O_3_-NiSe_2_-450°C-NC catalyst at various scanning speeds are shown in Appendix A. According to the relationship between the scanning rate square root (V_1/2_) of Nd_2_O_3_-NiSe_2_-450°C-NC and I_P_ (Appendix A), Nd_2_O_3_-NiSe_2_-450°C-NC has the largest electrochemical active region, and the oxidation–reduction process is primarily regulated by diffusion.

As contrast samples of Nd_2_O_3_-NiSe_2_-450°C-NC, we prepared neodymium oxide and nickel selenide. The UOR and MOR of the reference sample and the rare-earth-metal nickel-based catalyst were measured in pure KOH solution. The comparison revealed that all rare-earth-metal nickel-based catalysts exhibit redox peaks, indicating that the Nd_2_O_3_-NiSe_2_-450°C-NC catalyst has good catalytic activity and is successfully prepared.

The MOR behavior of the catalyst was evaluated via different test methods. CV tests were conducted in electrolytes with or without 0.5 M MeOH, as shown in Figure 4a. The results indicate that Nd_2_O_3_-NiSe_2_-450°C-NC exhibits a clear redox peak in the methanol-containing electrolyte. The LSV measurement of Nd_2_O_3_-NiSe_2_-450°C-NC shows that the potential of the catalyst is 1.33 V (up to 10 mA cm^−2^) after adding methanol (Figure 4b). This also shows that Nd_2_O_3_-NiSe_2_-450°C-NC has significant methanol oxidation performance. The results show that the temperature of selenidation has an effect on the MOR behavior of the catalyst, which has a potential of 1.33 V (up to 10 mA cm^−2^) after adding methanol, indicating that Nd_2_O_3_-NiSe_2_-450°C-NC has significant methanol oxidation performance. These findings demonstrate the effect of selenidation temperature on the MOR behavior of the catalyst. Nd_2_O_3_-Ni_3_Se_2_-350°C-NC, Nd_2_O_3_-NiSe_2_-450°C-NC, Nd_2_O_3_-NiSe_2_-550°C-NC, Nd_2_O_3_-NiSe-650°C-NC, and Nd_2_O_3_-NiSe-750°C-NC CV exhibit electrocatalytic activity in MOR (Figure 4c). Nd_2_O_3_-NiSe_2_-450°C-NC exhibits higher MOR activity than other catalysts because it has more active sites owing to the selenidation temperature (450 °C) and the rare-earth-element doping. The temperature of Nd_2_O_3_-Ni_3_Se_2_-350°C-NC and Nd_2_O_3_-NiSe-650°C-NC, Nd_2_O_3_-NiSe-750°C-NC after high- and low-temperature selenidation is found to be unsuitable, indicating that the temperature is not conducive to improving the catalyst’s electro-oxidation activity. Moreover, extremely high or extremely low selenide temperatures will destroy the structure of the catalyst. LSV tests on the catalysts obtained at various selenidation temperatures reveal that Nd_2_O_3_-NiSe_2_-450°C-NC has the lowest potential and best MOR performance (Figure 4d). Semicircles represent the spectra of all samples (Figure 4e), with the Nd_2_O_3_-NiSe_2_-450°C-NC catalyst having the smallest semicircle radius, indicating that the catalyst has high conductivity and charge transfer rate. At the same time, the charge transport impedance (R_ct_) of Nd_2_O_3_-NiSe_2_-450°C-NC is 5.5 Ω, lower than Nd_2_O_3_-Ni_3_Se_2_-350°C-NC (8 Ω), Nd_2_O_3_-NiSe_2_-550°C-NC (6 Ω), Nd_2_O_3_- NiSe-650 °C-NC (7.5 Ω), and Nd_2_O_3_- NiSe-750 °C-NC (17.5 Ω). The results show that the doping of rare-earth-element neodymium improves the conductivity and charge transfer rate of nickel-based catalysts at 450 °C.

In practical applications, durability is an important index to determine the service life of an electrocatalyst. The durability of the synthetic catalyst was tested for 4000 s using chronoamperometry at 1.5 V. The current density of all electrodes dropped sharply during the first few seconds. This decline could be because of the poisoning of the active site caused by the accumulation of strong-adsorption intermediate species. The current then gradually approached a steady state. Figure 4f shows that selenide catalysts at different temperatures have good durability, with the Nd_2_O_3_-NiSe_2_-450°C-NC catalyst having better durability than other catalysts. The Tafel slope of the catalysts with different selenidation temperatures to MOR was calculated by an LSV test. It can be seen from Appendix A that Nd_2_O_3_-NiSe_2_-450°C-NC has the smallest Tafel slope in the MOR test, which indicates that it has a fast kinetic process and that the catalyst is more prone to methanol oxidation reactions.

The results of the selenidation of the precursor at various temperatures revealed that selenidation at 450 °C improved the performance of methanol oxidation. The effect of doping on physicochemical properties at the ideal selenide temperature was then measured. According to the results obtained from the MOR performance test and the XRD, the only way to improve the catalyst activity is the production of NiSe_2_ via selenidation. Similarly, XPS revealed that the catalyst has potential oxygen vacancy due to neodymium oxide doping, and the creation of oxygen vacancy can potentially speed up charge transfer and increase conductivity. As a result of the LSV and CV tests, we can conclude that Nd_2_O_3_-NiSe_2_-450°C-NC has a higher starting potential and current density than NiSe_2_-450°C-NC and Nd_2_O_3_-NC (Appendix A). Nd_2_O_3_-NiSe_2_-450°C-NC exhibits higher electrocatalytic activity than other electrocatalysts at 450 °C selenide temperature.

The UOR behavior of the catalyst was examined in a variety of ways. The CV curve of Nd_2_O_3_-NiSe_2_-450°C-NC, before and after urea addition, is shown in Figure 5a. After urea addition, the catalyst exhibits an apparent redox peak. LSV testing was used to investigate the effect of urea on Nd_2_O_3_-NiSe_2_-450°C-NC potential. The results show that Nd_2_O_3_-NiSe_2_-450°C-NC has excellent UOR electro-oxidation activity, capable of reaching 10 mA cm^−2^ at 1.32 V (Figure 5b). Doping Nd_2_O_3_-NiSe_2_-450°C-NC with rare-earth elements exposes new active centers, increasing its UOR activity. In addition, the urea oxidation characteristics of Ni-based catalysts containing Nd prepared via selenidation at different temperatures were examined. The LSV curve (Figure 5c) and CV curve (Figure 5d) reveal that the UOR activity of Nd_2_O_3_-NiSe_2_-450°C-NC after different selenidation temperatures is higher than that of the catalyst. The oxidation kinetics of Nd_2_O_3_-NiSe_2_-450°C-NC in urea was faster than that of other selenide catalysts. All experimental results show that Nd_2_O_3_-NiSe_2_-450°C-NC has good catalytic activity. In addition, the electrocatalytic activity of Nd_2_O_3_-Ni_3_Se_2_-350°C-NC and Nd_2_O_3_-NiSe-750°C-NC in the process of urea oxidation is not very good. This experimental phenomenon is similar to that of methanol oxidation, indicating that high- and low-temperature selenidation are not conducive to improving the material’s electro-oxidation activity.

The effect of selenide temperature on the reaction was further studied. In the process of urea electrolysis, we concluded that Nd_2_O_3_-NiSe_2_-450°C-NC has the shortest semicircle radius (Figure 5e), showing that the charge transfer resistance of Nd_2_O_3_-NiSe_2_-450°C-NC is small and that the electrocatalysis kinetics of this catalyst is faster than that of other catalysts. The charge transport impedance (Rct) of Nd_2_O_3_-NiSe_2_-450°C-NC can also be obtained as 7 Ω, which is lower than Nd_2_O_3_-Ni_3_Se_2_-350°C-NC (9.8 Ω), Nd_2_O_3_-NiSe_2_-550°C-NC (7.3 Ω), Nd_2_O_3_-NiSe-650°C-NC (8.9 Ω), and Nd_2_O_3_-NiSe-750°C-NC (20.7 Ω). It also shows that the doping of rare-earth-element neodymium improves the catalytic performance of nickel-based catalysts [36,37]. In urea oxidation, the endurance of Nd_2_O_3_-NiSe_2_-450°C-NC was tested. Within 4000 s, the current density of Nd_2_O_3_-NiSe_2_-450°C-NC remained unaltered, showing that Nd_2_O_3_-NiSe_2_-450°C-NC has good durability (Figure 5f). However, the catalyst had obvious attenuation in the first 1000 s and remained unchanged after that. The attenuation was caused by the poisoning of products formed in the process of UOR [38,39]. The Tafel slopes of the catalysts at different selenidation temperatures were calculated by an LSV test in UOR. Appendix A shows that Nd_2_O_3_-NiSe_2_-450°C-NC has the smallest Tafel slope, which indicates that it has a fast kinetic process and that the urea oxidation reaction of the catalyst is more likely to occur. The LSV and CV test results indicate that the initial potential and current density of Nd_2_O_3_-NiSe_2_-450°C-NC are better than NiSe_2_-450°C-NC and Nd_2_O_3_-NC (Appendix A).

In comparison to other selenide temperature catalysts, the experimental investigation revealed that Nd_2_O_3_-NiSe_2_-450°C-NC is the most effective catalyst in alkaline methanol and urea electrolysis, and the reasons are the following: (1) the unique 3D nanosheet structure formed after selenidation can promote the electrocatalytic activity of the catalyst; (2) doping with selenide and carbon increases the electrochemical reaction activity and the electron transfer rate; and (3) at the same time, the synergistic effect of neodymium oxide and nickel selenide plays a catalytic role, improving the catalytic activity of UOR and MOR catalysts.

## 4. Conclusions

Nickel-based catalysts doped with rare-earth metals were successfully prepared by employing the salt-assisted template technique, selenidation method, and high-temperature pyrolysis method under low-cost and simple conditions. The catalyst was selenided at different temperatures, had high catalytic activity in alkaline methanol and urea electrolysis, and was an excellent electrocatalyst, as revealed by the physical characterization and electrochemical tests. The selenide and carbon doping increased the electrochemical reaction activity and electron transfer rate, but the unique design of Nd_2_O_3_-NiSe_2_-450°C-NC was largely responsible for its improved electrochemical properties. Furthermore, the synergistic action of neodymium oxide and nickel selenide helped in cocatalysts, improving the catalytic activity of UOR and MOR catalysts. Therefore, Nd_2_O_3_-NiSe_2_-450°C-NC can be a promising anode material.

## Figures and Tables

**Figure 1 nanomaterials-13-00814-f001:**
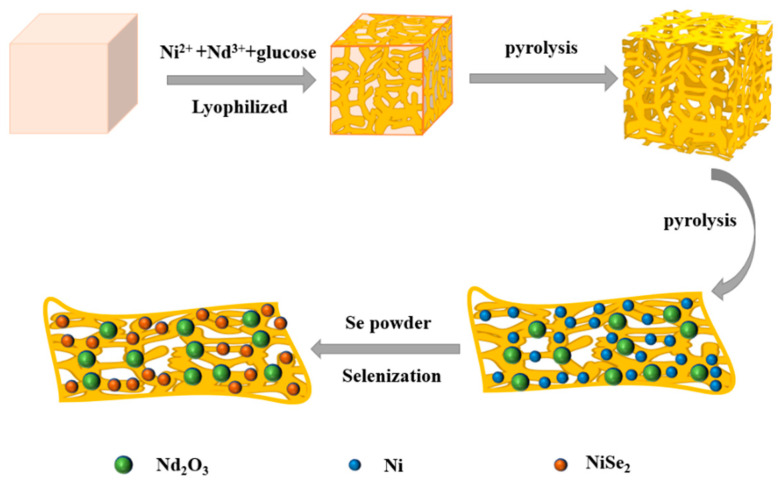
Synthesis diagrams of Nd_2_O_3_-NiSe_2_-450°C-NC.

**Figure 2 nanomaterials-13-00814-f002:**
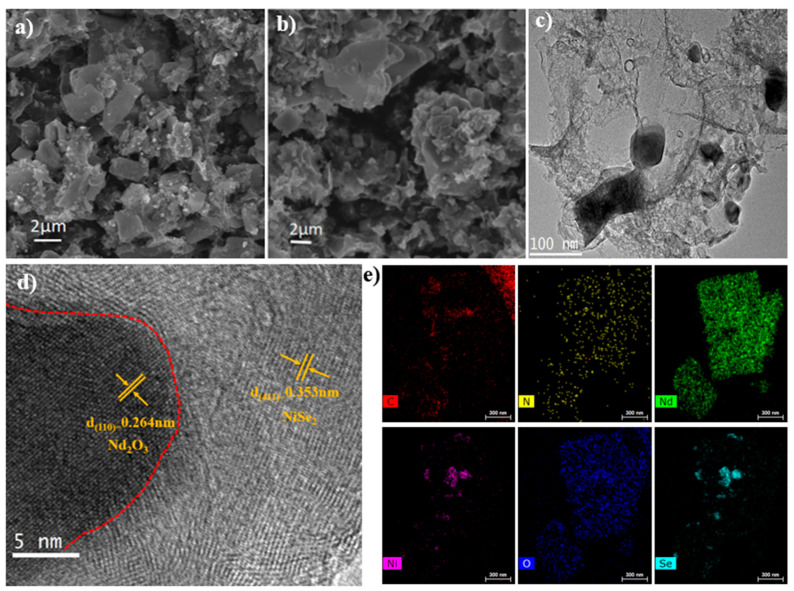
(**a**,**b**) SEM images of Nd_2_O_3_-NiSe_2_-450°C-NC at pre-selenide and post-selenide; (**c**) TEM images of Nd_2_O_3_-NiSe_2_-450°C-NC; (**d**) HR-TEM images of Nd_2_O_3_-NiSe_2_-450°C-NC; (**e**) elemental mapping of Nd_2_O_3_-NiSe_2_-450°C-NC (scale bar = 300 nm).

**Figure 3 nanomaterials-13-00814-f003:**
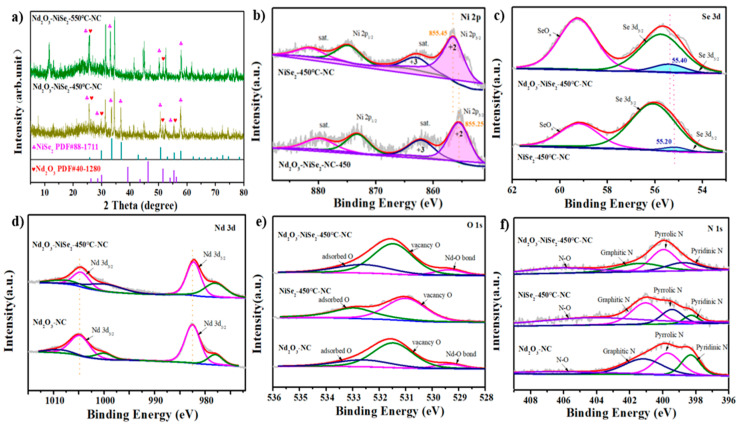
(**a**) XRD patterns of Nd_2_O_3_-NiSe_2_-450°C-NC and Nd_2_O_3_-NiSe_2_-550°C-NC; XPS spectrum tests of (**b**) Ni 2p, (**c**) Se 3d, (**d**) Nd 3d, (**e**) O 1s, and (**f**) N 1s of Nd_2_O_3_-NiSe_2_-450°C-NC, NiSe_2_-450°C-NC, and Nd_2_O_3_-NC.

**Figure 4 nanomaterials-13-00814-f004:**
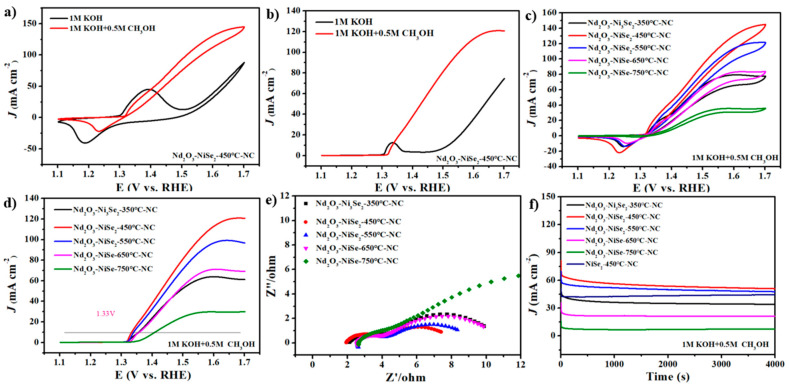
(**a**) CVs of Nd_2_O_3_-NiSe_2_-450°C-NC in 1 M KOH and 1 M KOH+ 0.5 M MeOH (sweep speed: 50 mV s^−1^); (**b**) LSV of Nd_2_O_3_-NiSe_2_-450°C-NC in 1M KOH and 1 M KOH+ 0.5 M MeOH; (**c**) CVs of Nd_2_O_3_-Ni_3_Se_2_-350°C-NC, Nd_2_O_3_-NiSe_2_-450°C-NC, Nd_2_O_3_-NiSe_2_-550°C-NC, Nd_2_O_3_-NiSe-650°C-NC, and Nd_2_O_3_-NiSe-750°C-NC in 1 M KOH containing 0.5 M MeOH (sweep speed: 50mV s^−1^); (**d**) LSV of Nd_2_O_3_-Ni_3_Se_2_-350°C-NC, Nd_2_O_3_-NiSe_2_-450°C-NC, Nd_2_O_3_-NiSe_2_-550°C-NC, Nd_2_O_3_-NiSe-650°C-NC, and Nd_2_O_3_-NiSe-750°C-NC in 1 M KOH containing 0.5 M MeOH; (**e**) EIS of Nd_2_O_3_-Ni_3_Se_2_-350°C-NC, Nd_2_O_3_-NiSe_2_-450°C-NC, Nd_2_O_3_-NiSe_2_-550°C-NC, Nd_2_O_3_-NiSe-650°C-NC, and Nd_2_O_3_-NiSe-750°C-NC in 1 M KOH containing 0.5 M MeOH; and (**f**) chronoamperometry curves Nd_2_O_3_-Ni_3_Se_2_-350°C-NC, Nd_2_O_3_-NiSe_2_-450°C-NC, Nd_2_O_3_-NiSe_2_-550°C-NC, Nd_2_O_3_-NiSe-650°C-NC, and Nd_2_O_3_-NiSe-750°C-NC in 1 M KOH containing 0.5 M MeOH at 1.5 V.

**Figure 5 nanomaterials-13-00814-f005:**
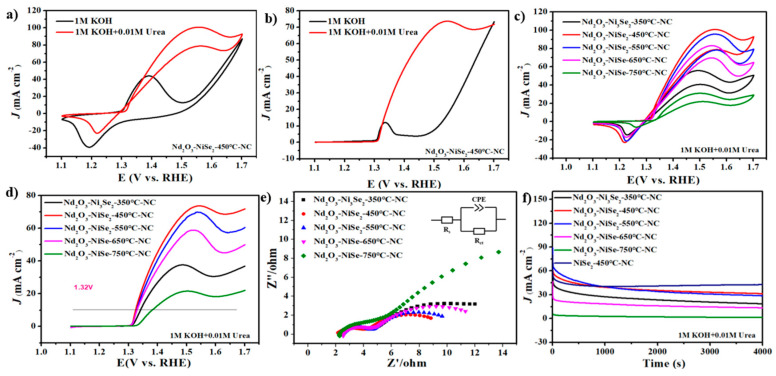
(**a**) CVs of Nd_2_O_3_-NiSe_2_-450°C-NC in 1 M KOH and 1 M KOH+0.01 M urea (sweep speed: 50 mV s^−1^); (**b**) LSV of Nd_2_O_3_-NiSe_2_-450°C-NC in 1 M KOH and 1 M KOH+0.01 M urea; (**c**) CVs of Nd_2_O_3_-Ni_3_Se_2_-350°C-NC, Nd_2_O_3_-NiSe_2_-450°C-NC, Nd_2_O_3_-NiSe_2_-550°C-NC, Nd_2_O_3_-NiSe-650°C-NC, and Nd_2_O_3_-NiSe-750°C-NC in 1 M KOH containing 0.01 M urea (sweep speed: 50 mV s^−1^); (**d**) LSV of Nd_2_O_3_-Ni_3_Se_2_-350°C-NC, Nd_2_O_3_-NiSe_2_-450°C-NC, Nd_2_O_3_-NiSe_2_-550°C-NC, Nd_2_O_3_-NiSe-650°C-NC, and Nd_2_O_3_-NiSe-750°C-NC in 1 M KOH containing 0.01 M Urea; (**e**) EIS of Nd_2_O_3_-Ni_3_Se_2_-350°C-NC, Nd_2_O_3_-NiSe_2_-450°C-NC, Nd_2_O_3_-NiSe_2_-550°C-NC, Nd_2_O_3_-NiSe-650°C-NC, and Nd_2_O_3_-NiSe-750°C-NC in 1 M KOH containing 0.01 M urea; and (**f**) chronoamperometry curves Nd_2_O_3_-Ni_3_Se_2_-350°C-NC, Nd_2_O_3_-NiSe_2_-450°C-NC, Nd_2_O_3_-NiSe_2_-550°C-NC, Nd_2_O_3_-NiSe-650°C-NC, and Nd_2_O_3_-NiSe-750°C-NC in 1 M KOH containing 0.01 M urea at 1.5 V.

## Data Availability

Not applicable.

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
