# Peer review of "Preparation of 3D Nd2O3-NiSe-Modified Nitrogen-Doped Carbon and Its Electrocatalytic Oxidation of Methanol and Urea"

_nanomaterials, 2023, doi:10.3390/nano13050814_

Round 1
Reviewer 1 Report
In this manuscript, the authors reported the Preparation of Nd2O3-NiSe-NC nanosheets and its electrocatalytic oxidation of methanol and urea. The topic of the manuscript is of scientific interest. The author provided sufficient data about the material characterization and electrochemical performance. Therefore, I suggest that the manuscript for published after addressing the following issues.
1. Structural information after electrochemical reactions should be provided.
2. MOR and UOR show a low potential to achieve high current densities (Figure 5a and b). For the high potentials, why they show a similar current density compared to the OER activities?
3. Some mistakes were found. Please carefully check and revised.
4. The following papers (Applied Catalysis B: Environmental, 2023, 325, 122388; Energy & Environmental Materials 2022, e12441 (DOI:10.1002/eem2.12441); Advanced Science, 2022, 9, 2204800) are recommended to be cited for improving the manuscript.
Author Response
1# In this manuscript, the authors reported the Preparation of Nd2O3-NiSe-NC nanosheets and its electrocatalytic oxidation of methanol and urea. The topic of the manuscript is of scientific interest. The author provided sufficient data about the material characterization and electrochemical performance. Therefore, I suggest that the manuscript for published after addressing the following issues.
1.Structural information after electrochemical reactions should be provided.
Response: Thank you very much for your comments and suggestions. Since the sample after electrochemical reaction cannot be tested before the beginning of the semester, we can observe the changes in the structure and composition of the catalyst after electrochemical characterization through the SEM and XPS diagrams of the catalyst. And from the literature we can find that the structure and composition of the catalyst after electrochemical characterization generally do not change.
[R1] Song, Y.; Ji, Z.; Zhao, S.; Wang, T.; Liu, J.; Hu, W., Reaction site exchange in hierarchical bimetallic Mn/Ni catalysts triggered by the electron pump effect to boost urea electrocatalytic oxidation. Journal of Materials Chemistry A 2022, 10 (19), 10417-10426.
[R2] Li, M.; Wu, X.; Liu, K.; Zhang, Y.; Jiang, X.; Sun, D.; Tang, Y.; Huang, K.; Fu, G., Nitrogen vacancies enriched Ce-doped Ni3N hierarchical nanosheets triggering highly efficient urea oxidation reaction in urea-assisted energy-saving electrolysis. Journal of Energy Chemistry 2022, 69, 506-515.
2.MOR and UOR show a low potential to achieve high current densities (Figure 5a and b). For the high potentials, why they show a similar current density compared to the OER activities?
Response: Following the reviewers' comments and suggestions. Compared to OER activities, MOR and UOR tests are more stable and subject to relatively little interference. Meanwhile, OER is a more complex process involving a four-electron mechanism, illustrating the slow surface kinetics of the electrocatalyst. Therefore, relatively speaking, the results of MOR and UOR tests can show similar current densities.
[R3]Chen J, Chen J, Cui H, Wang C. Electronic Structure and Crystalline Phase Dual Modulation via Anion-Cation Co-doping for Boosting Oxygen Evolution with Long-Term Stability Under Large Current Density. ACS Appl Mater Interfaces. Sep 25 2019;11(38):34819-34826.
[R4]Ding J, Han Y, Hong G. Tailoring the activity of NiFe layered double hydroxide with CeCO3OH as highly efficient water oxidation electrocatalyst. International Journal of Hydrogen Energy. 2021;46(2):2018-2025.
3.Some mistakes were found. Please carefully check and revised.
Response: We thank the reviewer for her/his comments and suggestions. I checked the article carefully and corrected the errors in the article. (Please see the revised manuscript)
4.The following papers (Applied Catalysis B: Environmental, 2023, 325, 122388; Energy&Environmental Materials 2022, e12441 (DOI:10.1002/eem2.12441); Advanced Science, 2022, 9, 2204800) are recommended to be cited for improving the manuscript.
Response: Following the reviewers' comments and suggestions,I have read the literature carefully and introduced it into my paper. (Please see the reference literature [10]、[18] and [24] )
[10] Sun H.; Xu X.; Kim H.; Jung W.; Zhou W.; Shao Z. Electrochemical Water Splitting: Bridging the Gaps between Fundamental Research and Industrial Applications. Energy & Environmental Materials. 2022.
[18] Sun H, Liu J, Kim H, et al. Ni-Doped CuO Nanoarrays Activate Urea Adsorption and Stabilizes Reaction Intermediates to Achieve High-Performance Urea Oxidation Catalysts. Advanced Science (Weinh). Dec 2022;9(34):e2204800.
[24] Sun H.; Li L.; Chen Y.; et al. Boosting ethanol oxidation by NiOOH-CuO nano-heterostructure for energy-saving hydrogen production and biomass upgrading. Applied Catalysis B: Environmental. 2023;325.

Reviewer 2 Report
The authors Zhang et al, reported Preparation of Nd2O3-NiSe-NC nanosheets and its electrocatalytic oxidation of methanol and urea. Although, this work contains some results, the organization and interpretation of the result should be enhanced further. Hence, I recommend this work required a substantial revision before considering for publications.
1. What is the theoretical potential for Urea oxidation?
2. Provide more experimental conditions in Figure 1.
3. Novelty of the work should be highlighted in the introduction in more clearly.
4. Author should include more results and data in the abstract.
5. Quality of images are poorly presented. It should be nicely presented.
6. The authors designed the work nicely, merely presented the results but failed to discuss the observed results elaborately.
7. Several typo errors and English language should be checked.
8. I suggest the authors to compare the previous literature similar to that work to find a merits of this work.
9. In order to strengthen the quality of the work, the following references should be cited; Renewable & Sustainable Energy Reviews, 143 (2021) 110849; Applied Catalysis B: Environmental 316 (2022) 121603.
Author Response
The authors Zhang et al, reported Preparation of Nd2O3-NiSe-NC nanosheets and its electrocatalytic oxidation of methanol and urea. Although, this work contains some results, the organization and interpretation of the result should be enhanced further. Hence, I recommend this work required a substantial revision before considering for publications.
1.What is the theoretical potential for Urea oxidation?
Response: The theoretical potential of urea oxidation was obtained by reviewing the literature.
* + CON2H4 à CON2H4*
CON2H4* + 2OH- àCO* + NH* + NH* +2H2O +2e-
CO* + NH* + NH* + 4OH- à * + * + * + CO2 + N2 + 4e- +4H2O
Thereinto * is the reaction interface of catalyst, CON2H4*, CO*, NH* are different intermediates adsorbed in the reaction process. For UOR, with the introduction of rare earth metals, the ∆G0 CON2H4* at the active site becomes more neutral, and the catalyst with more neutral adsorption energy has better catalytic performance, so it is observed that the catalyst UOR with the introduction of rare earth metals has better catalytic activity.
[R5] Wan H, Wang X, Tan L, et al. Electrochemical Synthesis of Urea: Co-reduction of Nitric Oxide and Carbon Monoxide. ACS Catalysis. 2023;13(3):1926-1933.
[R6] Wang G, Chen J, Li Y, et al. Energy-efficient electrolytic hydrogen production assisted by coupling urea oxidation with a pH-gradient concentration cell [J]. Chemical Communications, 2018, 54 (21): 2603-2606.
[R7] Xiao Z, Qian Y, Tan T, et al. Energy-saving hydrogen production by water splitting coupling Urea decomposition and oxidation reactions. Journal of Materials Chemistry A. 2023;11(1):259-267.
2.Provide more experimental conditions in Figure 1.
Response: Following the reviewers' comments and suggestions. The experimental steps and conditions in Figure 1 are reflected in the text. In this paper, dicyandiamide was added to the precursor using mixed freeze drying, salt template assisted technology and high temperature pyrolysis. The assisted pyrolysis of dicyandiamide was used to provide nitrogen source catalyst for preparation. It was selenized at different temperatures. (Please see Page 5)
3.Novelty of the work should be highlighted in the introduction in more clearly.
Response: Thank you very much for your comments and suggestions. I have added to the introduction in the text. (Please see the supplementary content in the introduction is marked in yellow)
4.Author should include more results and data in the abstract.
Response: We thank the reviewer for her/his comments and suggestions. More results and data are added to the summary . (Please see abstract content)
5.Quality of images are poorly presented. It should be nicely presented.
Response: Thank you very much for your comments and suggestions. The unclear images in the article are processed and revised. (Please see draft for revisions)
6.The authors designed the work nicely, merely presented the results but failed to discuss the observed results elaborately.
Response: Thank you very much for his correction. Some results of the paper are further analyzed. (Please see draft for revisions)
7.Several typo errors and English language should be checked.
Response: Thank the reviewer’s information. This article has been checked and some errors corrected. (Please see draft for revisions)
8.I suggest the authors to compare the previous literature similar to that work to find a merits of this work.
Response: Thank you very much for your comments and suggestions. In terms of experimental methods, calcination method was used in this experiment. Compared with other methods such as electrodeposition method and hydrothermal method, the process of catalyst preparation by this method is relatively more stable and safer. In the experimental results, compared with nickel selenide, the addition of rare earth neodymium can further improve the electrocatalytic activity of nickel selenide. Among rare earth elements, neodymium has the characteristics of large specific surface area, large nanometer size and high catalytic activity. It can promote the excellent catalytic performance of nickel-based catalyst.
[R8] Das A K, Pan U N, Sharma V, et al. Nanostructured CeO2/NiV-LDH composite for energy storage in asymmetric supercapacitor and as methanol oxidation electrocatalyst[J]. Chemical Engineering Journal, 2021, 417: 128019.
[R9] Abd El-Lateef H M, Almulhim N F, Mohamed I M A. Physicochemical and electrochemical investigations of an electrodeposited CeNi2@NiO nanomaterial as a novel anode electrocatalyst material for urea oxidation in alkaline media[J]. Journal of Molecular Liquids, 2020, 297: 111737.
[26] Ilanchezhiyan, P.; Mohan Kumar, G.; Siva, C.; Cho, H. D.; Lee, D. J.; Lakshmana Reddy, N.; Ramu, A. G.; Kang, T. W.; Kim, D. Y., Neodymium (Nd) based oxide perovskite nanostructures for photocatalytic and photoelectrochemical water splitting functions. Environmental Research 2021, 197, 111128.
9.In order to strengthen the quality of the work, the following references should be cited; Renewable & Sustainable Energy Reviews, 143 (2021) 110849; Applied Catalysis B: Environmental 316 (2022) 121603.
Response: We thank the reviewer for her/his comments and suggestions. I have read the literature carefully and introduced it into my paper. (Please see the reference literature [1] and [2])
[1] Lee SJ.; Theerthagiri J.; Nithyadharseni P.; et al. Heteroatom-doped graphene-based materials for sustainable energy applications: A review. Renewable and Sustainable Energy Reviews. 2021;143.
[2] Yu Y, Lee SJ.; Theerthagiri J.; Lee Y.; Choi MY. Architecting the AuPt alloys for hydrazine oxidation as an anolyte in fuel cell: Comparative analysis of hydrazine splitting and water splitting for energy-saving H2 generation. Applied Catalysis B: Environmental. 2022;316.

Reviewer 3 Report
The paper by M. Jia deals with the evaluation of the hybrid catalysts in the nanosheet form based on N-carbon containing Nd2O3-NiSe dopant prepared in electro oxidation of methanol and urea as a promising technology for hydrogen production instead more expensive HER process. The impact of catalysts composition adjusting and carbonization temperature on the electronic structure and catalytic performance enhancement of the synthesized materials was revealed. This paper may be helpful and interesting for the researchers specialized in the design of the functional electrocatalytic materials and green chemistry engineering. It may be published after minor revision by addressing the following points.
1. In Title. By definition, Nanosheets are 2D materials, not 3D.
2. Keywords should be supplemented by a couple of words, which highlight more strictly the paper’s main ideas.
3. The novelty of this work should be highlighted.
4. Please introduce the method of mixed freeze-drying.
5. The SEM and XRD studies should be presented more accurately. For instance, Line 134. This sentence is not clear: “The comparison diagram of these images revealed no considerable differences.”
6. Lines 146, 147. “The diffraction pattern of the material was analyzed by X-ray diffraction (XRD)”? “…The catalysts calcined at different temperatures could achieve the same peak value as Nd2O3”?
7. Lines 287-289. Please discuss in more quantitative way the catalyst activity.
Author Response
The paper by M. Jia deals with the evaluation of the hybrid catalysts in the nanosheet form based on N-carbon containing Nd2O3-NiSe dopant prepared in electro oxidation of methanol and urea as a promising technology for hydrogen production instead more expensive HER process. The impact of catalysts composition adjusting and carbonization temperature on the electronic structure and catalytic performance enhancement of the synthesized materials was revealed. This paper may be helpful and interesting for the researchers specialized in the design of the functional electrocatalytic materials and green chemistry engineering. It may be published after minor revision by addressing the following points.
1.In Title. By definition, Nanosheets are 2D materials, not 3D.
Response: Thank the reviewer’s information. Through SEM, we can observe that the catalyst presents a 3D nanosheet structure. (Please see Fig.2a and b in Page 7)
2.Keywords should be supplemented by a couple of words, which highlight more strictly the paper’s main ideas.
Response: We thank the reviewer for her/his comments and suggestions. Keywords are added to express the main ideas in the text. (Please see Page 2)
3.The novelty of this work should be highlighted.
Response: Following the reviewers' comments and suggestions. The novelty of the work is added in the introduction section. (Please see Introduction section)
4.Please introduce the method of mixed freeze-drying.
Response: We thank the reviewer for her/his comments and suggestions. Mixed freeze-drying is to put the prepared solution in the refrigerator and freeze it for one night, take it out and freeze it with liquid nitrogen for more than half an hour, put it in the freeze-drying machine for 24 hours, and freeze it into powder.
5.The SEM and XRD studies should be presented more accurately. For instance, Line 134. This sentence is not clear: “The comparison diagram of these images revealed no considerable differences.”
Response: Thank you very much for your comments and suggestions. The sentence in the passage was amended. (Please see Line 134)
6.Lines 146, 147. “The diffraction pattern of the material was analyzed by X-ray diffraction (XRD)”? “The catalysts calcined at different temperatures could achieve the same peak value as Nd2O3”?
Response: Thank you very much for his correction. Errors in this section have been corrected. (Please see Line 146-147)
7.Lines 287-289. Please discuss in more quantitative way the catalyst activity.
Response: We thank the reviewer for her/his comments and suggestions. According to LSV and CV tests, it can be concluded that Nd2O3-NiSe2-450℃-NC has higher starting potential and current density than NiSe2-450℃-NC and Nd2O3-NC. This also shows that Nd2O3-NiSe2-450℃-NC has MOR and UOR properties. (Please see Figure. S7b-c)
